# GENERATIVE DISCOVERY OF RELATIONAL MEDICAL ENTITY PAIRS

## ABSTRACT

Online healthcare services can provide the general public with ubiquitous access to medical knowledge and reduce the information access cost for both individuals and societies. To promote these benefits, it is desired to effectively expand the scale of high-quality yet novel relational medical entity pairs that embody rich medical knowledge in a structured form. To fulfill this goal, we introduce a generative model called *Conditional Relationship Variational Autoencoder* (CRVAE), which can discover meaningful and novel relational medical entity pairs without the requirement of additional external knowledge. Rather than discriminatively identifying the relationship between two given medical entities in a free-text corpus, we directly model and understand medical relationships from diversely expressed medical entity pairs. The proposed model introduces the generative modeling capacity of variational autoencoder to entity pairs, and has the ability to discover new relational medical entity pairs solely based on the existing entity pairs. Beside entity pairs, relationship-enhanced entity representations are obtained as another appealing benefit of the proposed method. Both quantitative and qualitative evaluations on real-world medical datasets demonstrate the effectiveness of the proposed method in generating relational medical entity pairs that are meaningful and novel.

## 1 INTRODUCTION

Increasingly, people engage in health services on the Internet (Fox & Duggan, 2013). The healthcare services can provide the general public with ubiquitous access to medical knowledge and reduce the information access cost significantly. The relational medical entity pair, which consists of two medical entities with a semantic connection between them, is an intuitive representation that distills human medical reasoning processes in a structured form. The medical relationships discussed in this paper are binary ones. For example, the Disease$\xrightarrow{Cause}$Symptom relationship indicates a "Cause" relationship from a disease entity to a symptom entity that is caused by this disease, such as the medical entity pairs $<$*Synovitis, Joint Pain*$>$. For the relationship Symptom$\xrightarrow{Belongto}$Department, we may have a relational medical entity pair such as $<$*Stiffness of a Joint, Orthopedics*$>$.

The ability to understand, reason and generalize is central to human intelligence (Oaksford & Chater, 2007). However, it possesses significant challenges for machines to understand and reason about the relationships between two entities (Santoro et al., 2017). Real-world relational medical entity pairs possess certain challenging properties to deal with: First, as the medical research develops, many medical relationships among medical entities that were once neglected due to the underdeveloped medical knowledge now need to be discovered. An increasing number of relationships will be formed among a large number of medical entities. Also, various linguistic expressions can be used for the same medical entity. For example, *Nose Plugged*, *Blocked Nose* and *Sinus Congestion* are symptom entities that share the same meaning but expressed very differently. Moreover, one medical relationship may instantiate entity pairs with varying granularities or relationship strength. For instance, Disease$\xrightarrow{Cause}$Symptom may include entity pairs like $<$*Rhinitis, Nose Plugged*$>$ as a coarse-grained entity pair, while $<$ *Acute Rhinitis, Nose Plugged*$>$, $<$*Chronic Rhinitis, Nose Plugged*$>$ are considered fine-grained entity pairs. As for the relationship strength, $<$*Cold, Fatigue*$>$ has greater relationship strength than $<$*Cold, Ear Infections*$>$ as cold rarely cause serious complications such as ear infections.

To effectively expand the scale of high-quality yet novel relational medical entity pairs, relation extraction methods (Culotta et al., 2006; Bach & Badaskar, 2007) are proposed to examine whether or not a semantic relationship exists between two given entities given a context. Although the existing relation extraction methods (Agichtein & Gravano, 2000; Baeza-Yates & Tiberi, 2007; Sahay et al., 2008; Yu & Lam, 2010; Chang et al., 2014; Wang et al., 2015) achieve decent performance in identifying the relationship for given entity pairs, those methods require contexts such as sentences retrieved from a large free-text corpus, from existing domain-specific knowledge graphs (Abacha & Zweigenbaum, 2011), or from web tables and links (Lin et al., 2010). As medical relationships in the real-world are becoming more and more complex and diversely expressed, existing relation extraction methods suffer from the data sparsity problem where it is hard to obtain additional external knowledge that covers all possible entity pairs, e.g. free-text corpus where two entities co-occur in the same sentence with a relationship between them. Therefore, it is crucial and appealing for us to discover meaningful relational medical entity pairs solely based on existing medical entity pairs, without the requirement of a well-maintained context as an additional external knowledge.

Furthermore, most relation extraction methods adopt a discriminative approach that learns to distinguish entity pairs of one relationship from the other (Zeng et al., 2014; Lin et al., 2016), or to identify meaningful entity pairs from randomly sampled negative entity pairs with no relationships (Bordes et al., 2013; Socher et al., 2013). Those methods need to iterate over the combination of all possible entity pairs and check each of them to discover new entity pairs. Such discriminative approach is tedious and labor-intensive. It is challenging yet rewarding for us to understand medical relationships intrinsically from the existing entity pairs. Specifically, in the medical domain, the diversely expressed medical entity pairs offer great advantages for us to ultimately understand medical relationships and discover high-quality relational medical entity pairs solely from existing meaningful medical entity pairs.

**Problem Studied:** We propose a novel research problem called RElational Medical Entity-pair DiscoverY (REMEDY), which aims at modeling relational medical entity pairs solely from the existing entity pairs. Also, it aims to discover meaningful and novel entity pairs pertaining to a certain medical relationship in a generative fashion, without sophisticated feature engineering and the requirement of external knowledge such as free-text corpora.

**Proposed Model:** A generative model named Conditional Relationship Variational Autoencoder (CRVAE) is introduced for relational medical entity pair discovery. It is unlikely to create meaningful, novel relational medical entity pairs without intrinsically understanding each medical relationship, more specifically, understanding the relationships between every two medical entities that instantiate a particular relationship. CRVAE fully explores the generative modeling capacity which roots in Bayesian inference while incorporating deep learning for powerful hands-free feature engineering. CRVAE is trained to encode each relational medical entity pair into a latent space conditioned on the relationship type. The encoding process addresses relationship-enhanced entity representations, interactions between entities as well as expressive latent variables. The latent variables are decoded to reconstruct entity pairs. Once the model is trained, we can sample directly from the distribution of latent variables and decode them into high-quality and novel relational medical entity pairs.

Overall, CRVAE has three notable strengths:

**CRVAE models the intrinsic relations between medical entity pairs directly** based on the existing meaningful relational medical entity pairs, without the requirement of additional external contexts for entity pair extraction. Existing relation extraction methods usually rely on the free-text corpus to decide whether a candidate entity pair it mentions is meaningful or not. The CRVAE only utilizes the existing entity pairs and pre-trained word vector as initial entity representations which are trained separately.

**CRVAE is able to generate entity pairs for a particular relationship**, even if it observes existing entity pairs only for that particular relationship. Unlike most discriminative methods which harness discrepancies among different relationships to distinguish the relationship of an entity pair from the other, or from randomly constructed negative entity pairs with no relations. The CRVAE understands the intrinsic medical relation from diversely expressed medical entity pairs and discovers meaningful, novel entity pairs of a particular relationship that we specified.

**CRVAE generates novel entity pairs** by a density-based sampling strategy in the generator. The generator samples directly from the latent space based on the density of hidden parameters. With the hands-free feature engineering by deep neural networks, the model is able to discover meaningful and novel entity pairs which does not exist in the training data.

The contributions of this paper can be summarized as follows:

- We study the Relational Medical Entity-pair Discovery (REMEDY) problem, which aims to expand the scale of high-quality yet novel relational medical entity pairs without maintaining large-scale context information such as the free-text corpus.
- We propose a generative model named Conditional Relationship Variational Autoencoder (CRVAE) that discovers relational medical entity pairs for a given relationship, solely from the diversely expressed entity pairs without sophisticated feature engineering.
- We obtain relationship-enhanced entity representations as an appealing benefit of the proposed model.

## 2 CONDITIONAL RELATIONSHIP VARIATIONAL AUTOENCODER

In this section, we introduce the Conditional Relationship Variational Autoencoder (CRVAE) model for the REMEDY problem. The proposed model consists of three modules: encoder, decoder, and generator. The encoder module takes relational medical entity pairs and a relationship indicator as the input, trained to intrinsically understand each relationship by translating and mapping the entity pair to a latent space as $Q_\phi$. The decoder is jointly trained to reconstruct the entity pairs as $P_\theta$. The generator model shares the same structure with the decoder, and it directly samples from the learned latent variable distribution to creatively generate meaningful medical relational entity pairs for a particular relationship. Figure 1 gives an overview of the proposed model.

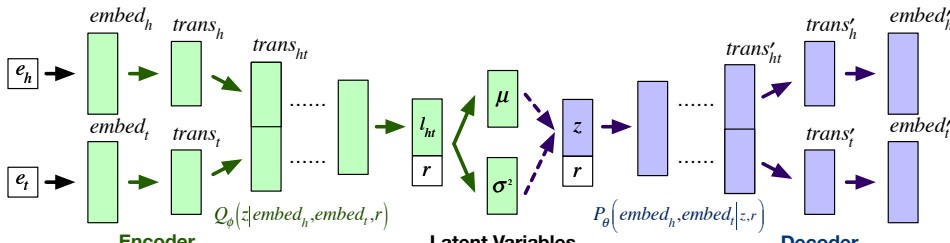

Figure 1: An overview of Conditional Relationship Variational Autoencoder (CRVAE) for Relational Medical Entity-pair Discovery during training. The encoder module is show in green color and the decoder module is show in blue. Model inputs are in white color.

The model takes a tuple $<e_h, e_t>$ and a relationship indicator $r$ as the input, where $e_h$ and $e_t$ are head and tail medical entity of a relationship $r$. For example, $e_h =$"*Synovitis*" and $e_t$="*Joint Pain*", while the corresponding $r$ is an indicator for Disease$\xrightarrow{Cause}$Symptom.

To effectively represent medical entities, pre-trained word embeddings that embody rich semantic information can be obtained as initial entity representations for $e_h$ and $e_t$. For simplicity, Skip-gram (Mikolov et al., 2013) is adopted to obtain 200-dimensional word embeddings trained separately and unsupervisely on a publicly accessible medical corpus. After a table lookup on the pre-trained word vector matrix $W_{embed} \in \mathbb{R}^{V \times D_E}$ where $V$ is the vocabulary size (usually tens of thousands) and $D_E$ is the dimension of the initial entity representation (usually tens or hundreds), $embed_h \in \mathbb{R}^{1 \times D_E}$ and $embed_t \in \mathbb{R}^{1 \times D_E}$ are derived as the initial embedding of medical entities.

### 2.1 ENCODER

With the initial entity representation $embed_h$ and $embed_t$ and their relationship indicator $r$, the encoder first translates and then maps entity pairs to a latent space as $Q_\phi(z|embed_h, embed_t, r)$.

### 2.1.1 TRANSLATING FOR RELATIONSHIP-ENHANCING

The initial embedding obtained from word embedding reflects semantic and categorical information. However, it is not specifically designed to model the medical relationship among medical entities (See observations in Section 3.4.3). To get entity representations that address relationship information, the encoder learns to translate each medical entity from its initial embedding to a relationship-enhanced embedding that distills relationship information. For example, a non-linear transformation can be used: $translate(x) = f(x \cdot W_{trans} + b_{trans})$ where $f$ can be an non-linear activation function such as the Exponential Linear Unit (ELU) (Clevert et al., 2015). $W_{trans} \in \mathbb{R}^{D_E \times D_R}$ is the weight variable and $b_{trans} \in \mathbb{R}^{1 \times D_R}$ is the bias where $D_R$ is the dimension for relationship-enhanced embeddings.

$$trans_h = translate(embed\_h), \quad trans_t = translate(embed\_t) \tag{1}$$

are obtained as relationship-enhanced embeddings for $e_h$ and $e_t$.

### 2.1.2 MAPPING TO LATENT VARIABLES

The relationship-enhanced entity representation $trans_h$ and $trans_t$ are concatenated $trans_{ht} = [trans_h, trans_t]$ and mapped to the latent space by multiple fully connected layers. For example, we can obtain a variable $l_{ht}$ that addresses the relationship information, as well as entity interactions from two medical entities, by applying three consecutive non-linear fully connected layers on $trans_{ht}$. As a variational inference model, we assume a simple Gaussian distribution of $Q_\phi(z|embed_h, embed_t, r)$ for the relational medical entity pair $<e_h, e_t>$ with a relationship $r$. Therefore, for each relational medical entity pair $<e_h, e_t>$ and a relationship indicator $r$, a mean vector $\mu$ and a variance vector $\sigma^2$ can be learned as latent variables to model $Q_\phi(z|embed_h, embed_t, r)$:

$$\mu = [l_{ht}, r] \cdot W_\mu + b_\mu, \quad \sigma^2 = [l_{ht}, r] \cdot W_\sigma + b_\sigma, \tag{2}$$

where a one-hot indicator $r \in \mathbb{R}^{1 \times |R|}$ is used for the medical relationship $r$ and $|R|$ is the number of all relationships. $W_\mu, W_\sigma \in \mathbb{R}^{(D_{l_{ht}} + |R|) \times D_L}$ are weight terms and $b_\mu, b_\sigma \in \mathbb{R}^{1 \times D_L}$ are bias terms. $D_L$ is the dimension for latent variables and $D_{l_{ht}}$ is the dimension for $l_{ht}$. To stabilize the training, we model the variation vector $\sigma^2$ by its log form $\log \sigma^2$ (to be explained in Equation 15).

## 2.2 DECODER

Once we obtain latent variables $\mu, \sigma^2$ for an input tuple $<e_h, e_t>$ which has the relationship $r$, the decoder uses latent variables and the relationship indicator $r$ to reconstruct the relational medical entity pair. The decoder implements the $P_\theta(embed_h, embed_t | z, r)$.

Given $\mu, \sigma^2$, it is intuitive to sample the latent value $z$ from the distribution $N(\mu, \sigma^2)$ directly. However, such operator is not differentiable thus optimization methods failed to calculate its gradient. To solve this problem, a reparameterization trick is introduced in Kingma & Welling (2014) to divert the non-differentiable part out of the network. Instead of directly sampling from $N(\mu, \sigma^2)$, we sample from a standard normal distribution $\epsilon \sim N(0, \mathrm{I})$ and then convert it back to $z$ by $z = \mu + \sigma\epsilon$. In this way, sampling from $\epsilon$ does not depend on the network.

Similarly as the use of multiple non-linear fully connected layers for the mapping in the encoder, multiple non-linear fully connected layers are used for an inverse mapping in the decoder. After the inverse mapping we obtain $trans'_{ht} \in \mathbb{R}^{1 \times 2D_R}$. The first $D_R$ dimensions of $trans'_{ht}$ are considered as a decoded relationship-enhanced embedding for $e_h$, while the last $D_R$ dimensions are for $e_t$:

$$trans'_h = trans'_{ht}[: D_R], \quad trans'_t = trans'_{ht}[D_R :], \tag{3}$$

where $trans'_h, trans'_t \in \mathbb{R}^{1 \times D_R}$. $trans'_h$ and $trans'_t$ are further inversely translated back to the initial embedding space $\mathbb{R}^{D_E}$:

$$embed'_h = f(trans'_h \cdot W_{trans\_inv} + b_{trans\_inv}), \quad embed'_t = f(trans'_t \cdot W_{trans\_inv} + b_{trans\_inv}),$$
$$\tag{4}$$

where $embed'_h, embed'_t \in \mathbb{R}^{1 \times D_E}$ are considered as reconstructed representations for $embed_h$ and $embed_t$.

## 2.3 TRAINING

Inspired by the loss function of the conditional variational autoencoder (CVAE) (Kingma et al., 2014; Sohn et al., 2015), the loss function of CRVAE is formulated to minimize the variational lower bound:

$$
\begin{aligned}
\mathcal{L}_{\text{CRVAE}}(embed_h, embed_t, r; \theta, \phi) = \\
- KL\left[Q_\phi\left(z|embed_h, embed_t, r\right)||P_\theta\left(z|r\right)\right] + \mathbb{E}\left[\log\left(P_\theta\left(embed_h, embed_t|z, r\right)\right)\right],
\end{aligned}
\tag{5}
$$

where $Q_\phi\left(z|embed_h, embed_t, r\right)$ is a simple Gaussian distribution used to approximate the unknown true distribution $P_\theta\left(z|embed_h, embed_t, r\right)$. $P_\theta\left(z|r\right)$ describes the true latent distribution $z$ given a certain relationship $r$ and $\mathbb{E}\left[\log\left(P_\theta\left(embed_h, embed_t|z, r\right)\right)\right]$ estimates the maximum likelihood.

A closed-form solution for the first term can be derived as:

$$
-\frac{1}{2}\sum_l^{D_L}\left(\exp\left(\sigma^2\right)_l + \mu_l^2 - 1 - \sigma_l^2\right),
\tag{6}
$$

where $\mu$ is the mean vector and $\sigma^2$ is the variance vector. $l$ in the subscript indicates the $l$-th dimension of the vector. Details for obtaining the closed-form solution are given in Appendix A

The second term penalizes the maximum likelihood, which is the conditional probability $P_\theta(embed_h, embed_t|z, r)$ of a certain entity pair $<e_h, e_t>$ given the latent variable $z$ and the relationship indicator $r$. The mean squared error (MSE) is adopted to calculate the difference between $<embed_h, embed_t>$ and $<embed'_h, embed'_t>$:

$$
\mathbb{E}\left[\log\left(P_\theta\left(embed_h, embed_t|z, r\right)\right)\right] = \frac{1}{2D_E}\left(||embed_h - embed'_h||_2^2 + ||embed_t - embed'_t||_2^2\right),
\tag{7}
$$

where $\|\cdot\|_2$ is the vector $\ell_2$ norm. To minimize the $\mathcal{L}_{\text{CRVAE}}$, existing optimizers such as Adadelta (Zeiler, 2012) can be used. Furthermore, a warm-up technique introduced in Sønderby & Raiko (2016) can let the training start with deterministic and gradually switch to variational, by multiplying $\beta$ to the first term. The final loss function used for training is formulated as:

$$
\mathcal{L}_{\text{CRVAE}} = -\frac{\beta}{2}\sum_l^{D_L}\left(\exp\left(\sigma^2\right)_l + \mu_l^2 - 1 - \log\sigma_l^2\right) + \frac{1}{2D_E}\left(||embed_h - embed'_h||_2^2 + ||embed_t - embed'_t||_2^2\right),
\tag{8}
$$

where $\beta$ is initialized as 0 and increase by 0.1 at the end of each training epoch, until it reaches 1.0 as its maximum.

## 2.4 GENERATOR

When we have a certain relationship $r$ in our mind that the generated relational medical entity pairs should belong to, a density-based sampling method is introduced for the generator to sample $\hat{z}$ from the latent space given a certain relationship $r$.

Instead of using the latent variable $z$ provided by certain $\mu$ and $\log\sigma^2$ in the encoding process from a certain $e_h, e_t$ and $r$, the generator tries to sample $\hat{z}$ directly from $P_\theta(\hat{z}|r)$ to get the latent space value $\hat{z}$ for a particular relationship $r$. Once $\hat{z}$ is obtained, the decoder structure is used to decode the relational medical entity pair. Figure 2 illustrates the generative process.

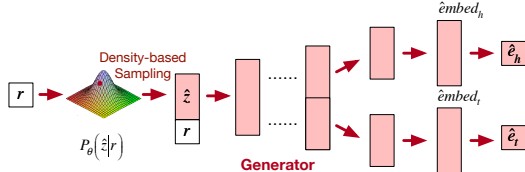

Figure 2: The generator that generate meaningful, novel relational medical entity pairs from the latent space.

The denser region in the latent space $P_\theta(\hat{z}|r)$ indicates that more densely entity pairs are located in the manifold. Therefore, a sampling method that considers the density distribution of $P_\theta(\hat{z}|r)$ samples more often from the denser regions in the latent space so as to preserve the

true latent space distribution of the sampled values. Specifically, for each relationship $r$, the density-based sampling samples $\hat{z}$ directly from $P_\theta(\hat{z}|r) \sim N(0, \mathrm{I})$, when trained properly. The resulting vectors $\hat{embed}_h$ and $\hat{embed}_t$ are mapped back to their entities in the initial embedding space $\mathbb{R}^{1 \times D_E}$, namely $\hat{e}_h$ and $\hat{e}_t$, by finding the nearest neighbor of the initial entity representation using $W_{embed}$. The $\ell$-2 distance measure is used for the nearest neighbor search.

# 3 EXPERIMENTS

## 3.1 DATASET & TRAINING DETAILS

The dataset consists of 46.02k real-world relational medical entity pairs in Chinese from a Chinese online healthcare forum www.xywy.com. The data set covers six different types of medical relationships. Table 1 shows the collection of relational medical entity pairs used in this study. 70% data are used for training and 30% for validation.

Table 1: Sample Medical Relationships and relational medical entity pairs.

| MEDICAL RELATIONSHIP | COUNT | RELATIONAL MEDICAL ENTITY PAIRS |
|---|---|---|
| Disease $\xrightarrow{Cause}$ Body Part | 2320 | < 三尖瓣闭锁(*tricuspid insufficiency*), 三尖瓣(*tricuspid valve*) > 
 < 阴道癌(*vaginal cancer*), 生殖(*reproductive system*) > 
 < 脑积水(*hydrocephaly*), 头部(*head*) > |
| Disease $\xrightarrow{RelatedTo}$ Disease | 4614 | < 婴儿脑积水(*infant hydrocephalus*), 先天性脑积水(*congenital hydrocephalus*) > 
 < 尿道炎(*urethritis*), 膀胱炎(*cystitis*) > 
 < 食滞胃脘(*retention of food in the stomach*), 小儿消化不良(*infantile indigestion*) > |
| Disease $\xrightarrow{Need}$ Examine | 4185 | < 水杨酸类中毒(*salicylates poisoning*), 尿常规(*routine urianlysis*) > 
 < 法洛三联症(*tetralogy triad*), 心电图(*electrocardiogram, ECG*) > 
 < 附睾炎(*epididymitis*) , 提睾反射(*cremasteric reflex*) > |
| Symptom $\xrightarrow{BelongTo}$ Department | 8595 | < 关节强直(*anchylosis, stiffness of a joint*), 骨科(*orthopedics*) > 
 < 女性小腹疼痛(*Female lower abdominal pain*), 妇科(*gynecology*) > 
 < 吸吮反射消失(*absent infant sucking reflex*), 新生儿科(*neonatology*) > |
| Disease $\xrightarrow{Cause}$ Symptom | 16642 | < 腹膜炎(*peritonitis*), 腹部静脉怒张(*abdominal venous engorgement*) > 
 < 尿道炎(*urethritis*), 尿道痒感(*urethra itching*) > 
 < 桡神经麻痹(*radial nerve palsy*), 上肢无力(*upper extremity weakness*) > |
| Symptom $\xrightarrow{RelatedTo}$ Symptom | 9662 | < 脐周红肿(*redness and swelling around the umbilicus*), 脐周肿胀(*periumbilical swelling*) > 
 < 肌肉挫伤(*muscular contusion*), 肌腱断裂(*disinsertion*) > 
 < 手指冻肿(*fingers benumbed with cold*), 皮肤冻伤(*skin frostbite*) > |

We use 200-dimensional word embeddings learned with the Skip-gram algorithm in Mikolov et al. (2013), trained from 6 million text corpus on the Chinese online healthcare forum as the initial entity representation. The vocabulary covers 126,270 words. We use Xavier initialization (Glorot & Bengio, 2010) for weight variables and zeros for biases. A wide range of hyperparameter configurations are tested with the proposed model. See Appendix B for detailed hyperparameter analysis.

## 3.2 PERFORMANCE EVALUATION

For each medical relationship, 1000 entity pairs are generated. Three evaluation metrics are introduced to quantitatively measure the generated relational medical entity pairs: quality, support, and novelty.

**Quality** Since it is hard for the machine to evaluate whether a relational medical entity pair is meaningful or not, human annotation is involved in assessing the quality of the generated relational medical entity pairs. A human annotation task is deployed on Amazon Mechanical Turk for annotation (Task shown in Appendix C). Similar as the precision metric adopted in Bach & Badaskar (2007), the quality[1] is measured by:

$$quality = \frac{\text{\# of entity pairs that are meaningful}}{\text{\# of all the generated entity pairs}}. \tag{9}$$

---

[1]Note that metrics such as recall is not applicable in such generative discovery task as the total population of positive samples is unknown.

**Support** Besides the quality metric, a support metric is developed to quantitatively measure the degree of belongingness of a generated entity pair to a relationship. For each generated relational medical entity pair $<\hat{e}_h, \hat{e}_t>$ and a candidate relationship $r_c$, the support score is calculated by:

$$support_{<\hat{e}_h, \hat{e}_t, r_c>} = \frac{1}{1 + distance(\hat{e}mbed_h, \hat{e}mbed_t)}, \tag{10}$$

where $distance(\hat{e}mbed_h, \hat{e}mbed_t)$ calculates the distance between the vector $\hat{e}mbed_h - \hat{e}mbed_t$ and $NN_{r_c}(\hat{e}mbed_h - \hat{e}mbed_t)$ using distance measure such as cosine distance. The $NN_{r_c}$ implements the nearest neighbor search over the $embed_h - embed_t$ space on all the training data which has the relationship $r_c$. For each generated medical entity pair, the support scores for all the candidate relationships are normalized so that they sum up to one:

$$norm\_support_{<\hat{e}_h, \hat{e}_t, r_c>} = \frac{support_{<\hat{e}_h, \hat{e}_t, r_c>}}{\sum\limits_{r_i}^{|R|} support_{<\hat{e}_h, \hat{e}_t, r_i>}}. \tag{11}$$

The relationship having the highest score is considered as the estimated relationship for $<\hat{e}_h, \hat{e}_t>$ while the relationship $r$ given during the generating process is considered as the ground truth for $<\hat{e}_h, \hat{e}_t>$. The final support value is based on the accuracy of the estimated relationship and the ground truth relationship.

**Novelty** The ability to generate novel relational medical entity pairs is one of our key contributions. Due to different scope of medical knowledge among individuals, human annotators are not able to precisely evaluate the novelty. We measure the novelty of the generation process by:

$$novelty = \frac{\text{\# of entity pairs that do not exist in the dataset}}{\text{\# of all the generated entity pairs}}. \tag{12}$$

### 3.3 BASELINES

Considering that no known methods are currently available for the REMEDY problem, we compare the performance of the following models:

- CRVAE-MONO: The proposed model which only takes one single type of relational medical entity pairs in both training and generation. For each type of relationship, we train a separate CRVAE only with entity pairs having that relationship.

- RVAE: The unconditional version of the model CRVAE where the relationship indicator $r$ is not provided during model training and generation.

- CRVAE-RAND: The proposed model CRVAE with a random sampling based generator. Unlike the density-based sampling adopted in CRVAE, the generator of CRVAE-RAND samples randomly from the latent space.

- CRVAE: The proposed method where relational medical entity pairs that belong to all types of relationships are used to train the model altogether. The training is conditioned on relationships and density-based sampling is used.

- CRVAE-WA: The proposed method with the warm-up strategy introduced in Section 2.3.

### 3.4 EXPERIMENT RESULTS

We summarize the performance of the proposed method, along with other alternatives, in Table 2.

CRVAE-MONO demonstrates the power of generative models in terms of learning the intrinsic representation and generating new entity pairs only given one type of relationship during the training (Quality: 0.6698, Support: 0.9550, Novelty: 0.5118). For CRVAE-RAND, although it generates highly novel (0.9952) entity pairs that are not seen in the training data, the generated entity pairs are of low quality (0.2550). By comparing CRVAE and CRVAE-RAND, we can see that the density-based sampling enables the generation of high-quality entity pairs that results in +47.58% in quality and +52.84% in support. The warm up technique adopted in CRVAE-WA is able to give CRVAE a further performance boost, where all measures improve consistently (+4.09% in quality, +2.43% in support and +5.11% in novelty).

Table 2: Performance of the proposed method with other baselines.

| MODEL NAME | QUALITY | SUPPORT | NOVELTY | LOSS (TRAIN / VALID) |
|---|---|---|---|---|
| CRVAE-MONO | 0.6698 | 0.9550 | 0.5118 | 47.3002 / 116.6739 |
| CRVAE-RAND | 0.2550 | 0.3764 | 0.9952 | 43.0954 / 83.6589 |
| CRVAE | 0.7308 | 0.9048 | 0.5682 | 43.0954 / 83.6589 |
| CRVAE-WA | 0.7717 | 0.9291 | 0.6193 | 33.4399 / 57.9470 |

As a qualitative measure, we also provide relational medical entity pairs generated by the proposed model. For example, the entity pair <痢疾(*dysentery*), 肠(*intestine*)> is generated given the medical relationship Disease$\xrightarrow{Cause}$Body Part, while entity pairs such as <阿米巴痢疾(*amebic dysentery*)，肠(*intestine*)> and <细菌性痢疾(*bacterial dysentery*), 胸部(*chest*)> are found in the training data. More entity pairs generated by the proposed method can be found in Appendix D.

### 3.4.1 GENERATIVE MODELING CAPABILITY

Unlike discriminative models which utilize the difference between instances of different classes to discriminate instances from one class to another, the proposed method purely learns from the existing relational medical entity pairs to generate new entity pairs. To validate such appealing property, Table 3 compares the fine-grained quality, support and novelty of entity pairs generated by CRVAE-MONO and CRVAE on each relationship.

Table 3: Quality, support and novelty metrics of the generated relational medical entity pairs by CRVAE-MONO and CRVAE.

| CRVAE-MONO | QUALITY | SUPPORT | NOVELTY | LOSS (TRAIN/VALID) |
|---|---|---|---|---|
| Disease$\xrightarrow{Cause}$Body Part | 0.683 | 1.000 | 0.488 | 54.9830 / 126.7426 |
| Disease$\xrightarrow{RelatedTo}$Disease | 0.689 | 0.870 | 0.483 | 51.5131 / 155.0721 |
| Disease$\xrightarrow{Need}$Examine | 0.708 | 1.000 | 0.521 | 54.7635 / 136.4802 |
| Symptom$\xrightarrow{BelongTo}$Department | 0.687 | 1.000 | 0.466 | 39.0959 / 72.5872 |
| Disease$\xrightarrow{Cause}$Symptom | 0.587 | 0.940 | 0.573 | 37.3276 / 83.8797 |
| Symptom$\xrightarrow{RelatedTo}$Symptom | 0.665 | 0.920 | 0.540 | 46.1180 / 125.2818 |
| **CRVAE** | | | | |
| Disease$\xrightarrow{Cause}$Body Part | 0.756 | 0.999 | 0.724 | |
| Disease$\xrightarrow{RelatedTo}$Disease | 0.691 | 0.744 | 0.867 | |
| Disease$\xrightarrow{Need}$Examine | 0.757 | 0.981 | 0.871 | 43.0954 / 83.6589 |
| Symptom$\xrightarrow{BelongTo}$Department | 0.768 | 0.995 | 0.613 | |
| Disease$\xrightarrow{Cause}$Symptom | 0.702 | 0.882 | 0.927 | |
| Symptom$\xrightarrow{RelatedTo}$Symptom | 0.711 | 0.828 | 0.888 | |

As shown in Table 3, the CRVAE-MONO on each relationship achieves a reasonable performance, which shows the capability of generative models in understanding every single medical relationship individually. Furthermore, when all types of entity pairs are trained and generated altogether in CRVAE, we observe a consistent improvement in not only quality but also novelty.

### 3.4.2 EFFECTIVENESS OF DENSITY-BASED SAMPLING

To validate the effectiveness of the density-based sampling for the generator, we compare the proposed method with CRVAE-RAND where a random sampling strategy is adopted. From Table 2 we can see that the random sampling strategy in CRVAE-RAND tends to generate more entity pairs that are not seen in the existing dataset. However, we observe a significant reduction in the quality

and support of the generated entity pairs when compared with CRVAE which adopts a density-based sampling. The dense region in the latent space indicates that more densely entity pairs are located. Therefore, in CRVAE, the quality and support of the generated entity pairs benefit from sampling more often at denser regions in the latent space, resulting in less novel but higher quality entity pairs.

### 3.4.3 EFFECTIVENESS OF RELATIONSHIP-ENHANCING ENTITY ADJUSTMENT

As mentioned in Section 2.1.1, the translating layer adjusts the original embedding to get relationship-enhanced entity representations. In the experiments, we study the embedding spaces before/after translation and observe that in the original embedding space, the Skip-gram tends to put entities that share similar context (e.g. *muscle strain* and *pull-up*) in proximity. While after relationship-enhancing, entities with similar functionalities in the same medical relationship are nearby with each other (e.g. *heart malformations* and *chromosome abnormalities*). See Appendix E for details.

### 3.4.4 ABILITY TO INFER CONDITIONALLY

One of our key contributions is that with proper training, the proposed method can generate relational medical entity pairs given a certain relationship. That is, the ability to infer new entity pairs for a particular relationship. Besides seamlessly incorporating this idea in the model design, we also visualize latent space of CRVAE and RVAE in order to show the conditional inference ability. See Appendix F for details.

## 4 RELATED WORKS

**Generative Models**: Recent years have witnessed an increasing interests in the research topic of generative models, which aims to generate observable data values based on some hidden parameters. Various generative models have been developed, such as Generative Adversarial Networks (GANs) (Goodfellow, 2016; Radford et al., 2015) and Variational Autoencoders (VAEs) (Kingma & Welling, 2013; Kingma et al., 2014; Sohn et al., 2015; Higgins et al., 2016; Nalisnick & Smyth, 2017). Unlike GANs which generate data based on arbitrary noises, the VAE setting adopted in this paper is more expressive for our task since it tries to model the underlying probability distribution of the data by latent variables so that new data from that distribution can be sampled accordingly.

There are some generative models and applications considering data in different modalities, such as generating images (Pu et al., 2016; Gregor et al., 2015; Dilokthanakul et al., 2016) or natural language texts (Bowman et al., 2016; Marcheggiani & Titov, 2016; Hu et al., 2017; Xu et al., 2017). As far as we know, the relational medical entity pair discovery problem we studied in this paper, which suits the generative purpose, has not been studied in a generative perspective.

**Relationship Extraction**: There is another related research area that studies relation extraction, which usually amounts to examining whether or not a relation exists between two given entities (Culotta et al., 2006). Most relationship extraction methods require large amounts of high-quality external information, such as a large text corpus (Baeza-Yates & Tiberi, 2007; Agichtein & Gravano, 2000; Sahay et al., 2008; Yu & Lam, 2010) and knowledge graphs (Wang et al., 2015; Chang et al., 2014; Syed et al., 2010). However, it is tedious and time-consuming to check each possible pair over all combinations of entities in the entity space. Thus, we propose an effective generative method that generates meaningful and novel relational medical entity pairs directly. Also, it is time consuming to collect and prepare a large corpus that covers all the mentions of those entity pairs, which makes it difficult to apply those methods. In this work, our model does not rely on additional external corpus for entity pair discovery.

Moreover, previous discriminative models usually need negative samples for supervised training. For example, Socher et al. (2013) trains the model to distinguish entity pairs with a relationship from randomly generated entity pairs as negative samples, while our model is can understand the medical relationship only from rational relational medical entity pairs thus even works when being fed with entity pairs having the same relationship type.

## 5 CONCLUSION

To effectively expand the scale of high-quality relational medical entity pairs which store the medical knowledge, a novel generative model named Conditional Relationship Variational Autoencoder (CRVAE) is introduced for Relational Medical Entity-pair Discovery (REMEDY). The proposed model fully explores the generative modeling ability while incorporates deep learning for powerful hands-free feature engineering. Unlike traditional relation extraction tasks which require additional contexts for extraction and need negative samples for discriminative training, the proposed method learns to intrinsically understand the medical relations from diversely expressed medical entity pairs, without the requirement of external context information. Moreover, it is able to generate meaningful, novel entity pairs for a given type of medical relationship. The relationship-enhanced entity representations have the potential to improve other NLP tasks. The performance of the proposed method is evaluated on real-world medical data both quantitatively and qualitatively.

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

# APPENDIX

## A DERIVING THE CLOSED-FORM SOLUTION

Inspired by the loss function of the conditional variational autoencoder (CVAE) Kingma et al. (2014); Sohn et al. (2015), the loss function of CRVAE is formulated to minimize the variational lower bound:

$$
\mathcal{L}_{CRVAE}(embed_h, embed_t, r; \theta, \phi) =
$$
$$
- KL\left[Q_\phi\left(z|embed_h, embed_t, r\right) || P_\theta\left(z|embed_h, embed_t, r\right)\right] + \log\left(P_\theta\left(embed_h, embed_t | r\right)\right), \tag{13}
$$

where the first term minimizes the KL divergence loss between the unknown true distribution $P_\theta\left(z|embed_h, embed_t, r\right)$ which is hard to sample from and a simple distribution $Q_\phi\left(z|embed_h, embed_t, r\right)$. The second term models the entity pairs by $\log\left(P_\theta\left(embed_h, embed_t | r\right)\right)$. The above equation can be reformulated as:

$$
\mathcal{L}_{CRVAE}(embed_h, embed_t, r; \theta, \phi) =
$$
$$
- KL\left[Q_\phi\left(z|embed_h, embed_t, r\right) || P_\theta\left(z|r\right)\right] + \mathbb{E}\left[\log\left(P_\theta\left(embed_h, embed_t | z, r\right)\right)\right], \tag{14}
$$

where $P_\theta\left(z|r\right)$ describes the true latent distribution $z$ given a certain relationship $r$ and $\mathbb{E}\left[\log\left(P_\theta\left(embed_h, embed_t | z, r\right)\right)\right]$ estimates the maximum likelihood. Since we want to sample from $P_\theta(z|r)$ in the generator, the first term aims to let to let $Q_\phi(z|embed_h, embed_t, r)$ to be as close as possible to $P_\theta(z|r)$ which has a simple distribution $N(0, \mathrm{I})$ so that it is easy to sample from. Furthermore, if $P_\theta(z|r) \sim N(0, \mathrm{I})$ and $Q(z|embed_h, embed_t, r) \sim N(\mu, \sigma^2)$, then a close-form solution for the first term can be derived as:

$$
-KL\left[Q_\phi\left(z|embed_h, embed_t, r\right) || P_\theta\left(z|r\right)\right] = -KL\left[N(\mu, \sigma) || N(0, \mathrm{I})\right]
$$
$$
= -\frac{1}{2}\left(tr\left(\sigma^2\right) + (\mu)^T\mu - D_L - \log\det\left(\sigma^2\right)\right) = -\frac{1}{2}\sum_l^{D_L}\left(\sigma_l^2 + \mu_l^2 - 1 - \log\sigma_l^2\right), \tag{15}
$$

where $l$ in the subscript indicates the $l$-th dimension of the vector. Since it is more stable to have exponential term than a log term, we model $\log\left(\sigma^2\right)$ as $\sigma^2$ which results in the final closed-form of Equation 15:

$$
-\frac{1}{2}\sum_l^{D_L}\left(\exp\left(\sigma^2\right)_l + \mu_l^2 - 1 - \sigma_l^2\right). \tag{16}
$$

## B HYPERPARAMETERS

We train the proposed model with a wide range of hyperparameter configurations, which are listed in Table 4. We vary the batch size from 64 to 256. The dimension $D_R$ for translating the initial entity embeddings is set from 64 to 2048. We try two to seven hidden layers from $trans_{ht}$ to $l_{ht}$ and from $[z, r]$ to $trans'_{ht}$, with different non-linear activation functions. For each hidden layer, the hidden unit number $D_H$ is set from 2 to 1024. The latent dimension $D_L$ is set from 2 to 200.

Table 4: Hyperparameter configurations.

| HYPERPARAMETER | VALUE |
|---|---|
| Batch Size | 64, 128, 256 |
| $D_R$ | 64, 128, 256, 512, 640, 768, 1024, 1280, 1536, 1792, 2048 |
| $D_H$ | 2, 4, 8, 16, 32, 64, 128, 256, 512, 640, 768, 1024 |
| $D_L$ | 2, 3, 4, 5, 10, 20, 50, 100, 200 |
| Activation | ELU (Clevert et al., 2015), ReLU (Nair & Hinton, 2010), Sigmoid, Tanh |
| Optimizer | Adadelta (Zeiler, 2012), Adagrad (Duchi et al., 2011), Adam (Kingma & Ba, 2014), RMSProp Tieleman & Hinton (2012) |

Table 5 shows the top five configurations ranked by their validation losses. From the combinations of those hyperparameter configurations, we find that for fully connected hidden layers from $trans_{ht}$ to $l_{ht}$, a sequence of six consecutive layers: 1792×640×640×512×256×64 works the best for the

Table 5: Model performance on different hyperparameter configurations. $\{D_H\}$ is a set of unit numbers for hidden layers in the encoder. For the decoder, hidden layers are organized in a reverse order.

| BATCH SIZE | $D_R$ | $\{D_H\}$ | $D_L$ | ACT. | OPTIMIZER | LOSS (TRAIN/VALID) |
|---|---|---|---|---|---|---|
| 64 | 640 | 1792×640×640×512×256× 64 | 200 | elu | adadelta | 43.0954 / 83.6589 |
| 64 | 640 | 1792×256×640×512×256×128 | 200 | elu | adadelta | 51.0695 / 86.9153 |
| 64 | 640 | 1792×256×640×512×256× 64 | 200 | elu | adadelta | 50.4392 / 88.6438 |
| 128 | 640 | 1792×640×768×512× 64×128 | 50 | elu | adadelta | 50.5997 / 89.0125 |
| 256 | 640 | 512×768×640×256×512 | 50 | elu | adam | 62.1955 / 89.2014 |

encoder with ELU as the activation function. For $[z, r]$ to $trans'_{ht}$ in the decoder, such layer setting is organized in a reverse order. A batch size of 64 and the Adadelta optimizer work the best for our task. $D_R = 640$ is used. The latent dimension $D_L = 200$ is adopted for $\mu$ and $\sigma^2$. Such configuration achieves a training loss of 43.0954 and a validation loss of 83.6589.

## C  TASK ON AMAZON MECHANICAL TURK

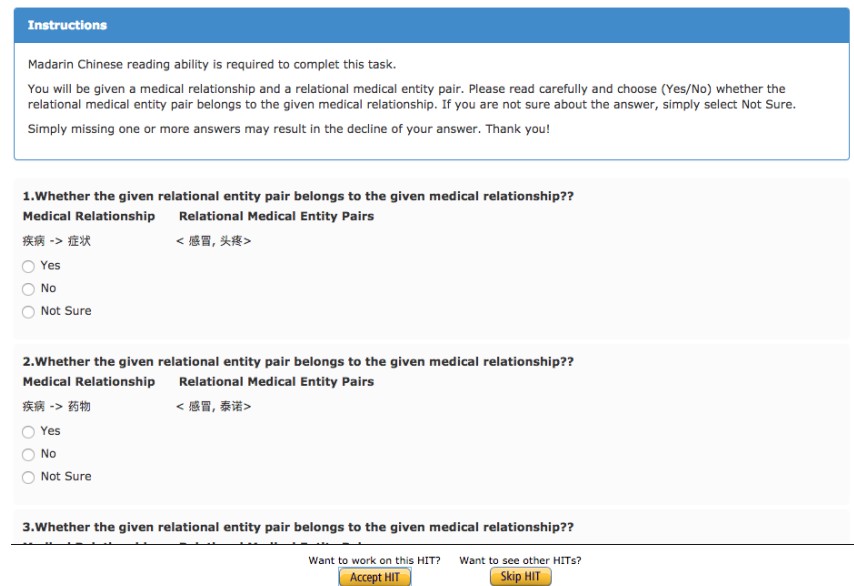

Figure 3: The screenshot of the human annotation task on Amazon Mechanical Turk.

## D  GENERATED RELATIONAL MEDICAL ENTITY PAIRS

Table 6 shows meaningful entity pairs generated by the proposed method.

## E  EFFECTIVENESS OF RELATIONSHIP-ENHANCING ENTITY ADJUSTMENT

To show the effectiveness of relationship-enhancement, Table 7 shows the nearest neighbors of a disease entity 生殖道畸形 (*genital tract malformation*) and a symptom entity 肌肉拉伤 (*muscle strain*) in their original embedding space, as well as the space after relationship-enhancing.

From these cases we can see that the original entity representations trained with Skip-gram (Mikolov et al., 2013) tend to put entities in proximity when they appear in similar contexts. In the first case, the entity 生殖道畸形(*genital tract malformation*) is in close proximity to 不孕 (*infertility*), 不孕症 (*acyesis*). In the second case, entities that have similar context like 引体向上 (*pull-up*) and 运动量 (*amount of exercise*) are found near by the entity 肌肉拉伤(*muscle strain*).

Table 6: Rational, meaningful relational medical entity pairs generated by the proposed method.

| |
|---|
| Disease $\xrightarrow{Cause}$ Body Part |
| < 痢疾(*dysentery*), 肠(*intestine*)> |
| < 脑瘤(*brain tumor*), 头部(*head*) > |
| < 白细胞减少症(*leukopenia*), 血液(*vascular system*) > |
| Disease $\xrightarrow{RelatedTo}$ Disease |
| < 食管异物(*foreign body in esophagus*), 肠梗阻(*bowel obstruction*) > |
| < 脑挫裂伤(*brain contusion*), 记忆障碍(*amnesia*) > |
| < 呼吸性酸中毒(*respiratory acidosis*), 肺水肿(*pulmonary edema*) > |
| Disease $\xrightarrow{Need}$ Examine |
| < 尿毒症(*uremia*), 尿常规(*routine urianlysis*) > |
| < 细菌性脑膜炎(*bacterial meningitis*), 头颅CT (*cranial CT*) > |
| < 肠梗阻(*bowel obstruction*), 腹部平片(*abdominal x-ray*) > |
| Symptom $\xrightarrow{BelongTo}$ Department |
| < 胎盘滞留(*retained placenta*), 产科(*obstetrics*) > |
| < 水潴留(*fluid retention*), 肾内科(*nephrology*) > |
| < 鼻塞(*stuffy nose*), 耳鼻咽喉科(*otolaryngology*) > |
| Disease $\xrightarrow{Cause}$ Symptom |
| < 耳源性脑脓肿(*otogenic brain abscess*), 耳痛(*earache*) > |
| < 神经炎(*neuritis*), 手麻(*numbness in the hands*) > |
| < 开放性颅脑损伤(*open head injury*), 意识模糊(*loss of consciousness*) > |
| Symptom $\xrightarrow{RelatedTo}$ Symptom |
| < 乏力(*fatigue*), 四肢无力(*feel wobbly and rough*) > |
| < 关节痛(*joint pain*), 关节活动受限(*limited joint mobility*) > |
| < 雾视(*blurred vision*), 眼睛不舒服(*eye discomfort*) > |

The translation layer adjusts the original entity representation so that they are more suitable for the Relational Medical Entity-pair Discovery task. The nearest neighbors in the adjusted space are not necessarily entities that co-occur in the same context, but more relation-wise similar with the given entity. For example, 心脏畸形 (*heart malformations*) and 染色体异常 (*chromosome abnormalities*) may not be semantically similar with the given word 生殖道畸形(*genital tract malformation*), but they may serve similar functionalities in a Disease $\xrightarrow{Cause}$ Symptom relationship.

## F  ABILITY TO INFER CONDITIONALLY

Figure 4 shows the values of validation data after being mapped into the $\mu$ space using RVAE (left) and CRVAE (right), respectively. The values are colored based on their ground truth relationship indicators. The left figure indicates that when the relationship indicator $r$ is not given during the training/validation, RVAE is still able to map different relationships into various regions in the latent space, while a single distribution models all types of relationships. Such property is appealing for an unsupervised model, but since the relationship indicator $r$ is not given, RVAE fails to generate entity pairs having a particular relationship, unless we manually assign a boundary for each relationship in the latent space. The right figure shows that when the relationship indicator $r$ is incorporated during the training, CRVAE learns to let each relationship have a unified latent representation. A separate but nearly identical distribution is used to model each medical relationship. Such property enables the generator of CRVAE to sample from a relationship-independent, unified latent space for diversities regarding the generation, while the relationship indicator $r$ given in CRVAE's generator provides categorical information on the type of relationship to generate.

Table 7: The effectiveness of relationship-enhancing adjustment on entity representations.

● 生殖道畸形(*genital tract malformation*)

| NN in the relationship-enhanced space $\mathbb{R}^{1 \times D_R}$ | NN in the initial embedding space $\mathbb{R}^{1 \times D_E}$ |
|---|---|
| 生殖道 (*genital tract*) | 生殖系统 (*reproductive system*) |
| 生殖系统 (*reproductive system*) | 生殖道肿瘤 (*reproductive tract tumors*) |
| 心脏畸形 (**heart malformations**) | 泌尿系畸形 (*urinary system malformations*) |
| 染色体异常 (**chromosome abnormalities**) | 不孕 (**infertility**) |
| 生殖道肿瘤 (*reproductive tract tumors*) | 阴道闭锁 (*vaginal atresia*) |
| 生殖器官 (*generative organs*) | 生殖道 (*genital tract*) |
| 泌尿系畸形 (**urinary system malformations**) | 生殖器官 (*generative organs*) |
| 消化道畸形 (**gastrointestinal malformations**) | 不孕症 (**acyesis**) |

● 肌肉拉伤(*muscle strain*)

| NN in the relationship-enhanced space $\mathbb{R}^{1 \times D_R}$ | NN in the initial embedding space $\mathbb{R}^{1 \times D_E}$ |
|---|---|
| 拉伤 (strain) | 拉伤 (*strain*) |
| 韧带拉伤 (**ligament strain**) | 肌肉撕裂 (*muscle tear*) |
| 扭伤 (*sprain*) | 引体向上 (**pull-up**) |
| 足痛 (**foot pain**) | 扭伤 (*sprain*) |
| 肌肉撕裂 (*muscle tear*) | 肌肉疲劳 (*muscle fatigue*) |
| 足底筋膜炎 (**plantar fasciitis**) | 腱鞘炎 (*tenosynovitis*) |
| 关节扭伤 (**joint sprain**) | 肌腱炎 (*tendonitis*) |
| 劳损 (*repetitive strain injury, RSI*) | 运动量 (**amount of exercise**) |

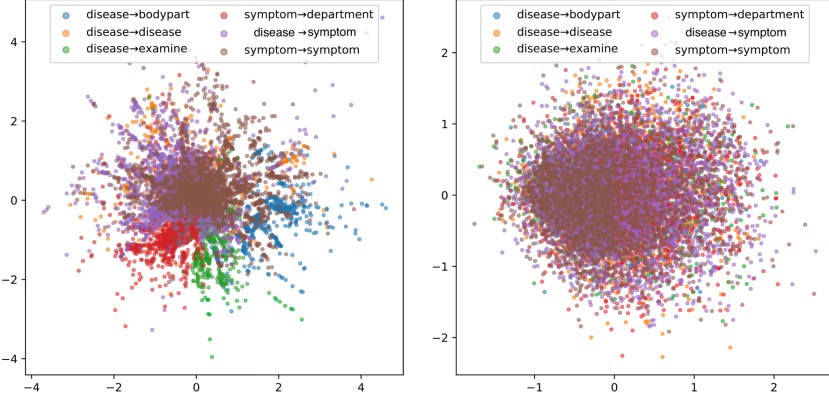

Figure 4: The latent variable $\mu$ of RVAE (left) and CRVAE (right) on the validation data, presented in a two-dimensional space after dimension reduction using Primary Component Analysis.

