# OpenReview forum: "Generative Discovery of Relational Medical Entity Pairs"
_ICLR.cc/2018/Conference — Reject_

### Official Review · AnonReviewer1 · 2017-11-22
**doubts on experimental setting**

**Rating:** 4
**Confidence:** 3

**Review:**

SUMMARY.

The paper presents a variational autoencoder for generating entity pairs given a relation in a medical setting.
The model strictly follows the standard VAE architecture with an encoder that takes as input an entity pair and a relation between the entities.
The encoder maps the input to a probabilistic latent space.
The latent variables plus a one-hot-encoding representation of the relation is used to reconstruct the input entities.
Finally, a generator is used to generate entity pairs give a relation.

----------

OVERALL JUDGMENT
The paper presents a clever use of VAEs for generating entity pairs conditioning on relations.
My main concern about the paper is that it seems that the authors have tuned the hyperparameters and tested on the same validation set.
If this is the case, all the analysis and results obtained are almost meaningless.
I suggest the authors make clear if they used the split training, validation, test.
Until then it is not possible to draw any conclusion from this work.

Assuming the experimental setting is correct, it is not clear to me the reason of having the representation of r (one-hot-vector of the relation) also in the decoding/generation part.
The hidden representation obtained by the encoder should already capture information about the relation.
Is there a specific reason for doing so?

---

> ### Author Response · Authors · 2018-01-05
> **Comments on Review**
>
> Thanks a lot for your review.
>
> 1.	The testing is not conducted on the validation set. The validation set is only used for hyperparameter tuning. We split the labeled entity pairs into training (70%) and validation (30%) set (described in the first paragraph of Section 3.1). As described in Appendix B, a hyperparameter analysis is conducted to show the validation losses when the model is trained with a wide range of hyperparameter settings, where the hyperparameter setting with the lowest validation loss is adopted.
> During testing, the proposed CRVAE model is able to generate unseen, meaningful entity pairs for a given medical relationship. The generator of the proposed model samples from the latent space according to the relationship of new entity pairs we want to obtain and then decodes the sampled vector, along with the relationship indicator, into entity pairs that are evaluated separately without the use of the validation set. The quantitative evaluation results are shown in Table 2 and Table 3, where three measurements are used: quality, support, and novelty. For qualitative evaluation, additional case studies and visualizations are provided in Section 3.4.2-3.4.4.
>
> 2.	We want to have a more controllable generation process in terms of which relationship of entity pairs we want to generate. The representation of r in the generation part enables the conditional inference: it guides the model to generate entity pairs having a certain relationship (instead of using a random noise to generate entity pairs having arbitrary relationships), which is one of our key contributions. As shown in Figure 2 in Section 2.4, the representation of r is fed to the generator in two stages: 1) when generating the latent vector $\hat z$ from the latent space 2) when decoding the sampled vector $\hat z$.
>
>         Another reason for introducing the representation of r into the generation process is that the latent space itself does not capture clear enough information without the use of the representation of r. We’ve introduced a baseline model RVAE (without incorporating r) and illustrated our observations in Figure 4. We color the labeled validation samples in the latent space, from which we can find that the baseline model RVAE (without incorporating r) is able to map entity pairs with different relationships vaguely into different regions in the latent space. However, since the label r is not used in RVAE, it is still hard to draw a clear enough boundary for each relationship, so as to sample accordingly and generate entity pairs having that relationship. This motivates us to incorporate the representation of r into the generation process. As shown in the right part of Figure 4, when r is given to the generator, the categorical information it provides naturally allows the generator to sample differently when the relationship varies. For example, if we want to generate entity pairs with symptom->disease relation, we will feed both the one-hot vector r indicating the symptom->disease relationship, as well as a latent value $\hat z$ sampled from the latent space that is conditioned on the symptom->disease relation, to the generator in order to get entity pairs having the symptom->disease relationship.

---

### Official Review · AnonReviewer3 · 2017-11-28
**An interesting application**

**Rating:** 4
**Confidence:** 4

**Review:**

The authors suggest using a variational autoencoder to infer binary relationships between medical entities. The model is quite simple and intuitive and the authors demonstrate that it can generate meaningful relationships between pairs of entities that were not observed before.
While the paper is very well-written I have certain concerns regarding the motivation, model, and evaluation methodology followed:

1) A stronger motivation for this model is required. Having a generative model for causal relationships between symptoms and diseases is "intriguing" yet I am really struggling with the motivation of getting such a model from word co-occurences in a medical corpus. I can totally buy the use of the proposed model as means to generate additional training data for a discriminative model used for information extraction but the authors need to do a better job at explaining the downstream applications of their model.

2) The word embeddings used seem to be sufficient to capture the "knowledge" included in the corpus. An ablation study of the impact of word embeddings on this model is required.

3) The authors do not describe how the data from xywy.com were annotated. Were they annotated by experts in the medical domain or random users?

4) The metric of quality is particularly ad-hoc. Meaningful relationships in a medical domain and evaluation using random amazon mechanical turk workers do not seem to go well together.

5) How does the proposed methods compare against a simple trained extractor? For instance one can automatically extract several linguistic features of the sentences two known related entities appeared with and learn how to extract data. The authors need to compare against such baselines or justify why they cannot be used.

---

> ### Author Response · Authors · 2018-01-05
> **Comments on Review**
>
> Thanks a lot for your review.
>
> 1.	In the medical domain, it is difficult to obtain a full spectrum of free-text in which all the relational medical entity pairs are co-occurred so that they can be further extracted in a discriminative fashion. The proposed generative method significantly lowers the data requirement for rational, novel medical entity pair discovery. It learns the intrinsic medical relations directly from the existing entity pairs without incorporating additional medical corpus in which two entities are co-occurred. As indicated in the review, the newly discovered entity pairs are definitely helpful in many ways: an intuitive downstream application is to provide more training samples for supervised learning models. Clustering could also benefit from the newly discovered entity pairs as a form of oversampling technique.
>
> 2.	We agree that the word embedding captures medical knowledge and embodies rich semantic information from the diversely expressed entity pairs. However, the word embedding cannot be removed for ablation study. It does not only build the backbone for entity pair representations and accelerate the model convergence, more importantly, the pre-trained word embeddings are necessary when decoding the generated word embeddings of the entity pairs into natural language entities. Without the word embedding, evaluation cannot be performed as we only obtain the generated embeddings, not entity pairs that are interpretable in the natural language for human annotation. Furthermore, the vocabulary of pre-trained word embedding is way larger than the number of unique entities in the labeled entity pairs. Using the word embedding may allow the model to decode unseen entities that exist in the vocabulary, but not in the training data.
>
> 3.	The relational medical entity pairs obtained from xywy.com are annotated manually by domain-experts.
>
> 4.	The generated relational medical entity pairs are evaluated both qualitatively and quantitatively. As far as we know, there is no existing quantitative metric for quality evaluation of the generated medical entity pairs. Therefore, human quality evaluation is conducted by Amazon Mechanical Turk workers. Instructions and requirements for workers are shown in Appendix C.
>
> 5.	The discriminative relation extraction from free-text and the generative entity pair discovery are two different tasks. The extractor is not explicitly evaluated in this work for the following reasons:
>         a.	Different training schema: the traditional extractor is trained discriminatively. It relies on the difference between entity pairs of different relationships and learns a decision boundary to distinguish one relation from another. The extractor fails to work in the case where all the training entity pairs belong to the same medical relation. Our generative setting solely learns from the existing entity pairs, no matter they belong to the same relationship or not. As shown in Table 3, our generative model works well when trained with entity pairs that all belong to the same relation, and works even better when entity pairs with different relations are trained together.
>         b.	Different testing schema: a large number of candidate entity pairs need to be provided and evaluated by the extractor in order to get the final, rational entity pairs. The choice of candidates sometimes involves additional expert knowledge; otherwise, any pairwise entities need to be fed to and tested by the extractor model. Our generative model learns to only generate rational medical entity pairs just given the type of relationship.  When testing, we genuinely believe that it is way more challenging to understand what an apple is in order to create a new apple with a different look, than simply trained to discriminate an apple from a banana. Thus it is unfair to simply compare their results.
>         c.	Use of data: Our model does not need external documents in both training/testing phase. It only requires labeled data and pre-trained word embeddings. The extractor suffers from the data sparsity problem during training: it is hard to obtain a full spectrum of documents where two medical entity pairs are not only mentioned simultaneously in a single sentence but also pertain a specific medical relationship in that sentence. Also, the extractor relies on keywords or indicators in a single sentence to determine the existence of a certain relation, which is not required by our model.

---

### Official Review · AnonReviewer2 · 2017-11-28
**Lack of advantages over (or evaluations against) pre-existing work**

**Rating:** 2
**Confidence:** 5

**Review:**

In the medical context, this paper describes the classic problem of "knowledge base completion" from structured data only (no text).  The authors argue for the advantages of a generative VAE approach (but without being convincing).  They do not cite the extensive literature on KB completion.  They present experimental results on their own data set, evaluating only against simpler baselines of their own VAE approach, not the pre-existing KB methods.

The authors seem unaware of a large literature on "knowledge base completion."  E.g. [Bordes, Weston, Collobert, Bengio, AAAI, 2011],  [Socher et al 2013 NIPS], [Wang, Wang, Guo 2015 IJCAI], [Gardner, Mitchell 2015 EMNLP], [Lin, Liu, Sun, Liu, Zhu AAAI 2015], [Neelakantan, Roth, McCallum 2015],

The paper claims that operating on pre-structured data only (without using text) is an advantage.  I don't find the argument convincing.  There are many methods that can operate on pre-structured data only, but also have the ability to incorporate text data when available, e.g. "universal schema" [Riedel et al, 2014].

The paper claims that "discriminative approaches" need to iterate over all possible entity pairs to make predictions.  In their generative approach they say they find outputs by "nearest neighbor search."  But the same efficient search is possible in many of the classic "discriminatively-trained" KB completion models also.

It is admirable that the authors use an interesting (and to my knowledge novel) data set.  But the method should also be evaluated on multiple now-standard data sets, such as FB15K-237 or NELL-995.  The method is evaluated only against their own VAE-based alternatives.  It should be evaluated against multiple other standard KB completion methods from the literature, such as Jason Weston's Trans-E, Richard Socher's Tensor Neural Nets, and Neelakantan's RNNs.

---

> ### Author Response · Authors · 2018-01-05
> **Comments on Review**
>
> Thanks for your review.
>
> 1.	The medical entity pairs generated by proposed model can be used to expand an existing knowledge graph with new entities as vertexes and relations as edges in a generative fashion. However, the KB completion task and the proposed entity pair discovery task share different objectives, and adopt totally different approaches:
>         a)	In the medical domain, it is difficult to obtain a full spectrum of free-text where all kinds of relational medical entity pairs are co-occurred. It is efficient to learn the intrinsic medical relations from existing entity pairs directly and generate unseen entity pairs in a generative fashion. Although both tasks provide additional entity pairs as the output, we approach this problem from a novel, generative perspective that significantly lowers the data requirements during training. Table 3 shows that our model works well even when all the training entity pairs have the same relationship. This can not be achieved by discriminatively trained KB completion methods. KB completion methods like Trans-E relies on entity pairs having different relations and learns to distinguish one from another; otherwise negative entity pairs with no semantics meanings are used.
>         b)	The generative discovery model is supposed to only generate rational entity pairs. Moreover, it is shown to have the ability to generate entity pairs having a pre-assigned relationship type, aka conditional inference, without the requirement of further domain knowledge. In the KB completion task, the rational entity pairs cannot be even obtained when there is no high-quality test set that contains entity pairs having that relationship. Otherwise, additional expert knowledge may be involved (e.g. to make sure that there exists a sentence that mentions two new entities having a certain relationship). Even then, the KB completion model needs to successfully classify the relationship for each test sample. The proposed model makes the conditional inference possible and efficient.
>         c)	Last but not least, it is unfair to simply evaluate the rational entity pairs generated by the proposed model against a discriminatively trained KB completion model that learns to tell the rational relation from other relations (or simply from a negative relation) when candidate entity pairs are already given for evaluation. We genuinely believe that it is way more challenging to understand what an apple is in order to create a new apple with a different look, than simply trained to distinguish an apple from a banana.
>
> 2.	In relation extraction methods where the objective is to detect whether or not a certain relation exists in a sentence, some words in the sentence serve as indicators. For example, for the "born in" relationship between a person and a place, words like "born", "from" are crucial. In the medical domain, free-text that contain a full spectrum of sentences that cover all medical entity pairs are hard to obtain, let alone domain-specific indicator words that are available to use. Without such text data as additional contexts, the proposed model is still able to generate novel entity pairs, which we consider as a major contribution.
>
> 3.	For our generative approach, "nearest neighbor search" is only performed as the last step of the decoder during evaluation to get natural language entities from the generated embeddings. Such operation is only performed on the generated rational entity pairs: it is not required at all during the training process. In many classic "discriminatively-trained" KB completion models, such search is usually used to trim candidate entity pairs that are not worth evaluating.
>
> 4.	The medical dataset has unique properties that other datasets do not have, which make it suitable for our generative entity pair discovery task.
> 	a)	First, the medical entity pairs contain clear and unambiguous relational semantics. This allows the model to directly encode two entities into the latent space without incorporating free-text contexts in which two medical entities are co-occurred. For example, the entity pairs <urethritis, urethra itching> and <radial nerve palsy, upper extremity weakness> can be used to learn the medical relationship from a disease to a symptom which it may cause. On the contrary, the entity pair <Obama, USA> in datasets, such as FB15K-237, possesses multiple relationships such as "born in", "president of", and "live in".
> 	b)	Second, different medical relationships used in this work are closely correlated with each other. For example, disease->disease, disease->symptom and symptom->symptom relationships share common entities, which is not frequently observed in other datasets. The proposed method is able to benefit from such property when solely learning from entity pairs. As shown in Table 3, quality and novelty are consistently improved when multiple correlated medical relationships are trained together, other than trained separately.

---

### Author Response · Authors · 2018-02-20
**CRVAE for Efficient Relational Modeling**

1) Comments on related works
Current discriminatively trained models share different objectives, and adopt entirely different approaches to discover novel entity pairs. They rely on context as external resources, or well-prepared candidate entity pairs for the models to examine. The proposed model significantly lowers the data requirement for efficient relational medical entity pair discovery:

Relation extraction methods usually require a substantial collection of contexts over a full spectrum of relationships that one wants to work on: e.g. contexts obtained from free-text corpora where two entities co-occur in the same sentence with a relationship between them. As medical relationships in the real-world are becoming more and more complex and diversely expressed, such context is hard to obtain.

Knowledge graph completion methods usually do not require contexts for training. However, they are vulnerable to the “garbage-in, garbage-out” situation during testing: we can not even obtain the rational medical entity pairs for a specific relationship when no high-quality entity pairs are having that relationship among the candidate entity pairs. The choice of candidates may involve additional human annotation; otherwise, any dyadic combinations of medical entities need to be fed to and tested by the model, which is tedious and labor-intensive. While the generative nature of our model makes it only generate rational entity pairs by learning from the existing rational ones: no additional data needs to be prepared for efficient generative discovery.


2) Comments on the experiment setting
The proposed model discovers entity pairs in a generative fashion: by directly sampling from the latent space, not by verifying pre-determined test cases. We don't need to prepare a test set for the model to examine. The validation set is not used for testing: it is used, and only used for hyperparameter study for the best model configuration. Quantitative and qualitative metrics such as Quality, Support, and Novelty are used to directly evaluate the meaningfulness and novelty of the generated entity pairs on real-world medical entity pairs.

---

### Decision · Program_Chairs · 2018-01-29
**ICLR 2018 Conference Acceptance Decision**

**Decision:**

Reject

**Comment:**

The authors seem to miss important related literature for their comparison.
They also tuned hyperparameters and tested on the same validation set.
They should split between train/validation/test.

Reviews are just too low across the board to accept.